# Effect of Cyclin-Dependent Kinase 4/6 Inhibitors on Circulating Cells in Patients with Metastatic Breast Cancer

**DOI:** 10.3390/cells13161391

**Published:** 2024-08-21

**Authors:** Soraia Lobo-Martins, Patrícia Corredeira, Ana Cavaco, Carolina Rodrigues, Paulina Piairo, Cláudia Lopes, Joana Fraga, Madalena Silva, Patrícia Alves, Lisiana Wachholz Szeneszi, Ana Barradas, Camila Castro Duran, Marília Antunes, Gonçalo Nogueira-Costa, Rita Sousa, Conceição Pinto, Leonor Ribeiro, Catarina Abreu, Sofia Torres, António Quintela, Gadea Mata, Diego Megías, Julie Ribot, Karine Serre, Sandra Casimiro, Bruno Silva-Santos, Lorena Diéguez, Luís Costa

**Affiliations:** 1Academic Trials Promoting Team, Institut Jules Bordet, Université Libre de Bruxelles (U.L.B.), 1070 Bruxelles, Belgium; soraialobomartins@gmail.com; 2Medical Oncology Department, Hospital de Santa Maria, Centro Hospitalar Universitário Lisboa Norte, 1649-028 Lisbon, Portugalleonor.ribeiro@ulssm.min-saude.pt (L.R.);; 3Instituto de Medicina Molecular João Lobo Antunes, 1649-028 Lisbon, Portugal; pcorredeira@medicina.ulisboa.pt (P.C.); acmcavaco@gmail.com (A.C.); scasimiro@medicina.ulisboa.pt (S.C.);; 4International Iberian Nanotechnology Laboratory, Avenida Mestre José Veiga s/n, 4715-330 Braga, Portugalclaudia_lopes96@outlook.com (C.L.);; 5RUBYnanomed Lda, Praça Conde de Agrolongo, 4700-314 Braga, Portugal; 6START Lisboa-CHULN Hospital Santa Maria, 1649-028 Lisbon, Portugal; patricia.alves@startlisbon.com; 7Centro de Estatística e Aplicações, Faculdade de Ciências, Universidade de Lisboa, 1749-016 Lisbon, Portugal; 8Faculdade de Medicina, Universidade de Lisboa, 1649-004 Lisbon, Portugal; 9Matemáticas y Computación Department, Universidad de La Rioja, 26006 Logroño, Spain; 10Confocal Microscopy Unit, Centro Nacional de Investigaciones Oncológicas (CNIO-ISCIII), 28029 Madrid, Spain; 11iMM Laço Hub, iMM-CARE, 1649-028 Lisbon, Portugal

**Keywords:** advanced breast cancer, CDK4/6 inhibitor, circulating tumor cell, ER+/HER2− breast cancer, immunomodulation, metastatic breast cancer, myeloid-derived suppressor cell, peripheral immune cell

## Abstract

The combination of cyclin-dependent kinase 4/6 inhibitors (CDK4/6i) with endocrine therapy (ET) is the standard-of-care for estrogen receptor (ER)-positive, HER2-negative (ER+/HER2− advanced/metastatic breast cancer (mBC). However, the impact of CDK4/6i on circulating immune cells and circulating tumor cells (CTCs) in patients receiving CDK4/6i and ET (CDK4/6i+ET) remains poorly understood. This was a prospective cohort study including 44 patients with ER+/HER2− mBC treated with CDK4/6i+ET in either first or second line. Peripheral blood samples were collected before (baseline) and 3 months (t2) after therapy. Immune cell’s subsets were quantified by flow cytometry, and microfluidic-captured CTCs were counted and classified according to the expression of cytokeratin and/or vimentin. Patients were categorized according to response as responders (progression-free survival [PFS] ≥ 6.0 months; 79.1%) and non-responders (PFS < 6.0 months; 20.9%). CDK4/6i+ET resulted in significant changes in the hematological parameters, including decreased hemoglobin levels and increased mean corpuscular volume, as well as reductions in neutrophil, eosinophil, and basophil counts. Specific immune cell subsets, such as early-stage myeloid-derived suppressor cells, central memory CD4+ T cells, and Vδ2+ T cells expressing NKG2D, decreased 3 months after CDK4/6i+ET. Additionally, correlations between the presence of CTCs and immune cell populations were observed, highlighting the interplay between immune dysfunction and tumor dissemination. This study provides insights into the immunomodulatory effects of CDK4/6i+ET, underscoring the importance of considering immune dynamics in the management of ER+/HER2− mBC.

## 1. Introduction

Advanced/metastatic breast cancer (mBC) remains a virtually incurable disease with a 5-year survival rate of approximately 38% [1]. In recent years, a notable advance in the standard-of-care for estrogen receptor-positive/human epidermal growth factor receptor 2-negative (ER+/HER2−) mBC has been the combination of cyclin-dependent kinase 4/6 inhibitors (CDK4/6i) with endocrine therapy (ET). The benefit of these agents extends to both first- and second-line settings, improving progression-free survival (PFS) and overall survival (OS) with manageable adverse events [2,3,4,5,6].

CDK4/6i inhibit the complex cyclin D1-CDK4/6, preventing the phosphorylation and subsequent inactivation of retinoblastoma protein. This blockade inhibits the release of E2F, leading to cell cycle arrest in the G1. Therefore, CDK4/6i play an important role in controlling cancer cell proliferation. CDK6 is particularly important in the differentiation of hematopoietic precursor cells [7], and thus the use of CDK4/6i may result in neutropenia. Unlike chemotherapy, this neutropenia is a consequence of cell cycle arrest and not apoptosis. Palbociclib or ribociclib, in contrast to abemaciclib, are associated with neutropenia of any grade in more than 70–80% of patients [2,3,4,5,6]. However, the impact of this cell arrest on hematological cells is not well understood. Further research is needed to clarify the effects of CDK4/6i on hematological cells and in the immune system.

It has been shown that unbalanced tumor-infiltrating immune cells within the tumor microenvironment (TME) can contribute to tumor growth and correlate with an unfavorable clinical outcome in many types of cancer, including breast cancer (BC). For example, tumor-infiltrating cytotoxic T cells (CD8+) and natural killer (NK) cells have been shown to promote an anti-tumor response [8,9], whereas total CD4+ T cells, regulatory T cells (Treg), and myeloid-derived suppressor cells (MDSCs) are considered pro-tumorigenic and suppress CD8+ T cells and NK cells [8,10].

Regarding the presence of these cells in the peripheral blood of BC patients, previous studies have reported a lower percentage of circulating CD4+ and CD8+ T cells compared to those infiltrating the TME of the same patients [11]. However, each T-cell subtype (CD4+, CD8+, Vδ1+, and Vδ2+) comprises a variety of subpopulations, including effector (Eff), effector memory (EM), central memory (CM), and naïve cells, with specific functions. Table 1 summarizes the immunophenotypic signatures and major functions of the various T-cell subtypes investigated in this study, highlighting the diverse roles and phenotypic characteristics of these cells within the immune system. 

Circulating tumor cells (CTCs) are shed into the bloodstream, leaving the primary tumor or metastatic sites, with the capacity to evade and colonize new organs. Therefore, CTC detection holds promise for the real-time monitoring of the tumor progression [30].

The aim of this study was to characterize changes in circulating immune cell subsets in mBC patients treated with CDK4/6i plus ET (CDK4/6i+ET) and to investigate how these changes are associated with clinical outcomes.

## 2. Materials and Methods

### 2.1. Study Design and Eligibility Criteria

This study is a subgroup analysis of patients enrolled in the ONCODYNAMICS BioBanking (ODB) project, which aims to support precision medicine in cancer by evaluating tumor clonal evolution, host immune response, and circulating biomarkers. From July 2017 to April 2023, patients with ER+/HER2− mBC were enrolled in a unicentric prospective cohort. All patients were treated with CDK4/6i in either first- or second-line in combination with an aromatase inhibitor (AI) or fulvestrant until disease progression or unacceptable toxicity. Pre- or peri-menopausal women underwent ovarian function suppression (OFS) with a gonadotropin-releasing hormone (GnRH) agonist. Demographic, clinical, and pathological data were collected. Clinical staging was performed at baseline using contrast-enhanced computed tomography (CT) and, if clinically indicated, magnetic resonance imaging or positron emission tomography (PET). Clinical benefit and objective tumor response were assessed by the medical doctor according to RECIST 1.1 criteria every 3–4 months or as clinically indicated [31]. Additional details on the inclusion and exclusion criteria for this sub-analysis are described in the Appendix A.

Patients were categorized as responders (Resp, PFS ≥ 6.0 months) or non-responders (NResp, PFS < 6.0 months) to CDK4/6i+ET, using the European Society for Medical Oncology Clinical Practice Guideline definition of primary resistance to ET for mBC [1]. 

Peripheral blood samples were collected in ethylene diamine tetra acetic acid before (baseline) and 3 months after (t2) therapy initiation. Analytical blood findings were retrieved from medical records initiation.

### 2.2. Peripheral Blood Mononuclear Cell Isolation and Immune Cell Characterization

Peripheral blood mononuclear cells (PBMCs) were isolated from patients’ whole blood on the day of collection. Each blood sample was mixed 1:1 vol/vol with phosphate-buffered saline (PBS) 1× and Histopaque-1077 Hybri-Max (Sigma-Aldrich, Irvine, UK), followed by centrifugation at 800× *g* for 30 min with brake set to off. The PMBC ring was carefully transferred to a new tube and mixed with RPMI-1640 (supplemented with 10% fetal bovine serum [Gibco^TM,^, ThermoFisher scientific, Burkington, ON, Canada], 5% penicillin-streptomycin [Gibco^TM^], 1% HEPES [Gibco^TM^], 1% sodium pyruvate [Gibco^TM^], and 1% MEM-NEAA [Gibco^TM^]). 

Immune cell populations were identified as previously described [21]. Gating strategy is depicted in Appendix A. To quantify B cells, T cells, NK cells, natural killer T (NKT) cells, and myeloid lineage cells by flow cytometry, a standard protocol for CD3, CD14, CD16, and CD19 staining was used. Vδ1, Vδ2, CD4, CD8, FOXP3, and CD45RA expression was used to differentiate total γδ T cells (Vδ1+, Vδ2+), αβ T cells (CD4+, CD8+), and Treg (subtypes I, II, and III) cells. Staining for CD27 and CD45RA was performed to identify effector (Eff), effector memory (EM), central memory (CM), and naïve populations of γδ T and αβT cells. Cytokine-positive cells (TNF-α, IFN-y, and IL-17) were also quantified after standard 3 h stimulation with 2 mg/mL BFA (Enzo Life Sciences Inc., Farmingdale, NY, USA), 40 µg/mL PMA (Enzo Life Sciences Inc.), and 80 µg/mL ionomycin (Enzo Biochem Inc., Farmingdale, NY, USA).

The anti-human fluorescently labeled mAbs used are indicated in Appendix A.

Samples were acquired on a BD LSR-Fortessa 2 (BD Biosciences, Milpitas, CA, USA) with daily quality control monitored with Sphero™ Rainbow Calibration Particles (8 peaks) (BD Biosciences). Analysis was performed using FlowJo^TM^ v10.6.2 software. 

### 2.3. Serum IFN-γ Levels

Serum IFN-γ levels were measured in the baseline sample of 32 patients by ELISA (Invitrogen^TM^, Bender MedSystems GmbH, Vienna, Austria) according to the manufacturer’s instructions. 

### 2.4. CTC Isolation and Characterization

A sample of 7.5 mL whole blood was injected at 160 μL/min into a dedicated microfluidic device designed to isolate CTCs based on size and deformability (Patent: PCT/EP2016/078406), as previously described [32,33]. The retained cells were washed with PBS 1× and fixed with formalin 4%, permeabilized with 0.25% Triton in PBS 1×, blocked with 2% BSA in PBS 1×, and stained with an antibody cocktail (mouse monoclonal anti-human pan cytokeratin FITC (1:200, C11, Sigma-Aldrich); mouse monoclonal anti-human vimentin eFluor 570 (1:50, V9, eBioscience™) and mouse monoclonal anti-human CD45 Alexa Fluor 647 (1:50, 35-Z6, Santa Cruz Biotechnology, Inc., Dallas, TX, USA), containing DAPI (10 μg/mL), for 1 h at room temperature. Each cell was imaged under fluorescence microscopy at 20× magnification for phenotypical analysis. First, a cell location and segmentation script was implemented to obtain the data used in the cell classifier plugin. Next, an automated CTC classifier was developed in-house and employed to classify isolated cells (for research use only) to maximize standardization. This implemented framework combined the automated image analysis of fluorescence microscopy images with human expert revision and validation. CTCs were identified as DAPI+/CD45− and CK+ or Vim+ or CK+/Vim+ cells, as previously described [33,34].

### 2.5. Statistical Analysis

Descriptive statistical analysis was performed. The Mann–Whitney–Wilcoxon test was used to compare the percentage of each immune cell population, IFN-γ serum levels, blood counts, and tumor markers (CEA, CA15.3, LDH) between Resp and NResp patients at baseline.

The point-biserial correlation was used to assess the correlation between the percentage of immune populations, IFN-γ serum levels, blood counts, tumor markers (CEA, CA15.3, LDH), and the presence/absence of CTCs [35]. A very strong correlation was considered when 0.8 < r ≤ 1, a strong correlation when 0.6 < r ≤ 0.8, and a moderate correlation when 0.4 < r ≤ 0.6.

A significance level of 0.05 was considered appropriate for all analyses. All variables analyzed are depicted in Appendix A.

Statistical analysis was performed using R (version 4.2.1) and RStudio (version 2023.12.1+402) using the following R packages, data.table, ggplot2, ggpubr and dplyr.

### 2.6. Ethics

This study was conducted in accordance with the tenets of the Declaration of Helsinki, good clinical practice, and local regulations. The protocol was approved by the Ethics Committee of Centro Hospitalar Universitário Lisboa Norte (Ethical Committee approval no 343). Written informed consent was obtained from all participating patients.

## 3. Results

A total of 44 patients were eligible for statistical analysis; 34 (77.3%) were classified as Resp and 10 (22.7%) as NResp. The median follow-up for the entire cohort was 38.07 months (95% confidence interval [CI] 30.26–41.74). 

### 3.1. Baseline Characteristics of the Study Population

The baseline clinical and pathological characteristics of the study cohort population are summarized in Table 2. All patients were female, with a median (range) age at study entry of 56.5 (27–88) years. More than half of the patients were post-menopausal at baseline (n = 28, 63.6%), and the majority of patients were treated in the first-line setting (n = 33, 75.0%). Non-special type (NST) carcinoma was the most common histologic sub-type (72.7%), with 90.0% of tumors being intermediate-high grade, 59.1% having high Ki67 (≥20%), and almost two-thirds having no HER2 expression (HER2 0, 65.9%). More than 70% of patients had disease-free survival (DFS) > 24 months, with 18.2% having de novo metastatic disease.

Both the Resp and NResp groups were well balanced with respect to demographic and clinicopathological characteristics (Table 1). Although more patients in the NResp compared to the Resp group presented with bone-only disease at baseline, this difference was not statistically significant (60.0% vs. 32.4%, *p* = 0.114). 

The analysis of baseline blood samples showed that Resp patients exhibited significantly lower levels of carcinoembryonic antigen (CEA) (median CEA 3.4 vs. 11.0 ng/mL for Resp and NResp, respectively; *p* = 0.0077, Figure 1A); and higher basophil counts (median basophil count 0.02 vs. 0.03/L for Resp and NResp, respectively; *p* = 0.0054, Figure 1B).

Regarding peripheral immune cell populations, Resp patients had significantly more effector (CD45RA+ CD27−) Vδ2 T cells than NResp patients at baseline (*p* = 0.0394, Figure 1C).

In Appendix A, all *p*-values are depicted for all analyzed variables.

### 3.2. Immune Cell Populations Correlate with the Presence of CTCs

The point-biserial correlation was used to evaluate the relationship between the presence (≥1) or absence (0) of CTCs at baseline and circulating immune cell subsets, blood parameters, and IFN-γ serum levels. A subset of 22 patients was eligible for analysis (Figure 2). 

The presence of CTCs (≥1) at baseline showed a strong negative correlation with effector CD8+ T cells (r = −0.69, *p* = 0.0004; Figure 2A) and a moderate negative correlation with Vδ1 T cells (r = −0.47, *p* = 0.0287; Figure 2B), effector Vδ1 T cells (r = −0.46, *p* = 0.0310; Figure 2C), and NKG2D-expressing Vδ1 cells (r = −0.47, *p* = 0.0263; Figure 2D). Noteworthy, a moderate negative correlation was found with the effector/total memory CD8 T-cell ratio (r = −0.44, *p* = 0.0414; Figure 2E) and the effector/central memory CD8 T-cell ratio (r = −0.48, *p* = 0.0242; Figure 2F). In contrast, a strong positive correlation was found for central memory CD8 T cells (r = 0.68, *p* = 0.0005; Figure 2G). In addition, a moderate positive correlation was also observed in naïve and central memory Vδ1 T cells (r = −0.47, *p* = 0.0287; Figure 2H,I).

Higher serum IFN−γ levels were negatively correlated with the presence of CTCs (r = −0.43, *p* = 0.0447; Figure 2J).

In Appendix A, all *p*−values are depicted for all analyzed variables.

### 3.3. Impact of CDK4/6i Treatment Plus ET on Immune Populations

Next, we aimed to assess the impact of therapy on analytical blood findings and immune cell populations, by analyzing paired blood samples collected at baseline and after 3 months of the treatment beginning (t2) (n = 23).

Regarding whole blood count (WBC), at t2, there was a notable decrease in hemoglobin (Hb) levels (median Hb 13.1 and 11.9g/dL at baseline and t2, respectively; *p* = 0.0003, Figure 3A) and a concurrent increase in mean corpuscular volume (MCV) (median 89.3 and 94.7fL at baseline and t2, respectively; *p* = 0.0008, Figure 3B). Additionally, at the 3-month timepoint, reductions in leucocyte, neutrophil, eosinophil, and monocyte counts were observed (leucocytes: median 5.92 and 3.20 × 10^9^/L at baseline and t2, respectively; *p* < 0.0001, Figure 3C; neutrophils: median 3.16 and 1.34 × 10^9^/L at baseline and t2, respectively; *p* = 0.0001, Figure 3D; eosinophils: median 0.11 and 0.05 × 10^9^/L at baseline and t2, respectively; *p* = 0.0043, Figure 3E; monocytes: median 0.33 and 0.23 × 10^9^/L at baseline and t2, respectively; *p* = 0.0130, Figure 3F).

In Appendix A, all *p*-values are depicted for all analyzed variables.

Furthermore, changes in circulating immune cell populations were also highlighted in the analysis. A decrease was observed in early-stage MDSCs (eMDSCs; *p* < 0.001; Figure 4A), central memory CD4+ T cells (*p* = 0.0078; Figure 4B), and Vδ2 T cells expressing NKG2D (*p* = 0.0122) after 3 months of treatment (Figure 4C).

### 3.4. Variation in Immune Cell Subsets According to Response to CDK4/6i Plus ET

To understand how WBC and immune cell subsets varied with CDK4/6i+ET and if this variation was dependent from clinical response to therapy, paired baseline and t2 samples were compared between Resp (n = 20) and NResp (n = 3) patients.

After 3 months of treatment, lower levels of eMDSCs were found in the NResp patients (*p* = 0.0087, Figure 5A), accompanied by leukocytosis (*p* = 0.0466, Figure 5B) and basophilia (*p* = 0.0142, Figure 5C).

Additionally, to evaluate the immune dynamics between t2 and baseline in the Resp and NResp patients, the baseline value was subtracted from the t2 value in both groups. There was a significant difference between baseline and t2 for the Treg III subset, which was higher in NResp patients (*p* = 0.0124, Figure 6A), indicating a decrease at t2 in NResp patients. Although not statistically significant, Treg I decreased in Resp patients at t2 (*p* = 0.0556; Figure 6B), making the difference between the two time points less than in NResp patients. These changes were accompanied by an increase in the CD8+ T cells expressing NKG2D in Resp (*p* = 0.0294; Figure 6C).

In Appendix A, all *p*-values are depicted for all analyzed variables.

## 4. Discussion

The present study aimed to identify the alterations in circulating immune cell subsets in patients with ER+/HER2− mBC undergoing CDK4/6i+ET and to investigate their association with clinical benefit, namely PFS. 

Baseline characteristics were well balanced between the therapy responder and non-responder groups. Notably, responders had higher baseline basophil counts and a greater presence of effector Vδ2 T cells (Vδ2+ CD45RA+ CD27−) compared to non-responders. 

Additionally, significant changes were identified in peripheral blood immune cell populations following CDK4/6i+ET. A decrease in neutrophil, eosinophil, and basophil counts was observed, suggesting the occurrence of hematological changes consistent with CDK4/6 inhibition. This finding is consistent with previous reports [3,36,37,38,39,40]. Changes in the specific immune cell subsets, namely effector Vδ2+ T cells, were also observed, further contributing to the understanding of the immunomodulatory effects of CDK4/6i+ET. In addition, correlations between the presence of CTCs and immune cell subsets at baseline were observed, shedding light on the interplay between immune dysfunction and tumor progression in mBC. 

Despite the well-established prognostic significance of various clinical parameters in ER+/HER2− mBC, this study disclosed intriguing findings. Typically, patients with unfavorable prognostic factors—such as endocrine resistance, the presence of visceral metastases, previous use of chemotherapy, or higher ECOG-PS—would be expected to have a poorer treatment response and prognosis [41,42]. However, the absence of significant imbalances in these parameters between therapy responders and non-responders in this cohort suggests potential underlying mechanisms influencing the treatment response beyond conventional prognostic factors.

Human γδ T cells may have a dual function in cancer, both contributing to carcinogenesis and participating in anti-tumor immunity. The Vδ2+ population is heterogeneous and capable of producing pro-inflammatory cytokines, such as TNF-α, IFN-γ, IL-17, IL-9, and IL-10. To the best of the authors’ knowledge, the impact of effector Vδ2+ T cells on mBC outcomes under CDK4/6i+ET was not previously investigated. Zoledronic acid has been reported triggering the activation and proliferation of Vδ2+ T cells, thereby inhibiting cancer cell growth [19,21]. However, all patients in this study only received denosumab as a bone-targeted agent (BTA), making this an unexpected increase. Mariani et al. showed that non-responders have a higher frequency of effector Vδ T cells, while responders have a lower frequency. Even when considering total Vδ T cells and acknowledging that Vδ2+ T cells are the most common γδ T-cell subtype in circulation [22], the results of this study are consistent with the findings reported in Mariani’s study. A low frequency of effector Vδ2+ T cells has been reported in glioblastoma [43]. In addition, relapsed patients with acute lymphoblastic leukemia tend to have a lower frequency of effector Vδ2+ T cells compared to disease-free patients [18].

CDK4/6i induce a dysregulation of the immune system in ER+/HER2− mBC patients, which has been extensively described by several authors [16]. The polymorphonuclear myeloid-derived suppressor cells (PMN-MDSCs; CD16− CD14− CD11b+ CD33+) play an important role in immune inhibition in the cancer microenvironment, while their role in circulation is still unclear. In the present study, a significant decrease in this population was observed three months after initiation of CDK4/6i+ET. This finding supports the previously described hypothesis of immunosuppression induced by this therapy [19,44]. In contrast, in 51 patients with mBC, a reduction in total MDSCs was observed without treatment specificity [45].

A decrease in central memory CD4+ T cells was observed 3 months after starting CDK4/6i+ET. Memory cells are a heterogeneous population with a long lifespan and immunologic memory [28]. A decrease in the central memory subset may be due to conversion to effector memory and then effector T cells. In the patients from the RIBECCA trial, no significant difference in central memory CD4+ T cells was found [46]. In our study, a reduction in Vδ2+ T cells expressing NKG2D was found after three months of CDK4/6i+ET. NKG2D is a lectin-like receptor type-2 gene expressed by γδ T, CD8+ T, NK, and NKT cells, allowing their recognition of the tumor and triggering their cytotoxic response [23,47]. This suggests that a reduction observed three months after the start of treatment may reflect a stabilization of the population within the immune response.

The correlation between the presence and absence of CTCs at baseline with immune populations and blood parameters was also investigated in this study. Because CTCs are continuously shed from the tumor into the bloodstream and have a very short half-life, they reflect the phenotype and genotype, as well as the heterogeneity of the tumor of origin. As such, it is relevant to understand how they may be related to clinical outcomes, changes in host immune cells, and disease progression, increasingly moving towards precision medicine [44,48]. 

The presence of CTCs was positively correlated with Vδ2 naïve T cells and central memory Vδ1+ and CD8+ T cells in this study. While naïve T cells have no effector function, central memory T cells have a higher proliferative capacity and induce differentiation into effector cells [26,27,49]. A negative correlation was observed in effector (CD8+ and Vδ1 T cells), Vδ1+ expressing NKG2D and Vδ1 total T cells. This finding could be explained by the accumulation of CTCs due to less anti-tumor activity [50]. 

IFN-γ is a well-known effector cytokine mainly produced by T and NK cells [51]. In this study, a negative correlation was found between IFN-γ and the presence of at least one CTC, in agreement with another study [52]. 

Several studies have classified FoxP3+ Treg cells into three subpopulations: Treg I (CD45RA+ FoxP3^lo^), Treg II (CD45RA− FoxP3^hi^), and Treg III (CD45RA− FoxP3^lo^) [46,47]. This study’s findings have shown a slight decrease in the Treg I subpopulation in responders, which was associated with response and good prognosis. These cells remain in a resting state until stimulation and subsequent activation. Treg III cells, also known as non-activated Treg cells, have reduced immunosuppressive activity in circulation [46,47]. In this cohort, an increase in Treg III cells was observed from baseline to t2 in patients responding to CDK4/6i+ET.

The present study has several limitations that should be acknowledged. First, the heterogeneity within the study population, including variations in endocrine sensitivity and lines of treatment (including both first- and second-line setting), may have introduced confounding factors with a potential impact on result interpretation. Despite our efforts to mitigate this through a prospective cohort design and accepting patient referrals to increase sample size, variability remains inherent in real-world evidence studies and not all patients underwent a second blood draw. The reasons for this included disease progression before the scheduled second collection, patient refusal, and disruptions caused by the COVID-19 pandemic. Consequently, the lack of follow-up blood samples may have limited the ability to capture longitudinal changes in immune cell populations and investigate their association with treatment response. Furthermore, the relatively small cohort size may have affected the statistical power and generalizability of the study findings. These constraints, frequent in academic initiatives, underscore the need for larger, more homogeneous patient cohorts and careful data collection protocols in future studies to draw more robust conclusions.

In conclusion, this study provides valuable insights into the effects of CDK4/6i+ET therapy on immune cell dynamics in patients with ER+/HER2− mBC. Despite baseline heterogeneity, CDK4/6i+ET induced significant alterations in hematological parameters and immune cell subsets. The findings of this study also suggest a complex interplay between immune dysfunction and tumor dissemination, as evidenced by correlations between the presence of CTCs and specific immune cell populations.

## Figures and Tables

**Figure 1 cells-13-01391-f001:**
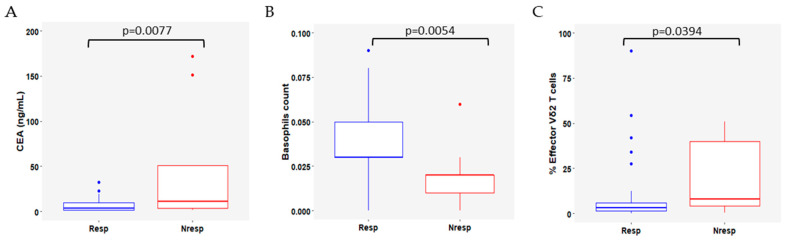
Significant differences at baseline between responders (Resp, blue) and non-responders (NResp, red) patients for (**A**) carcinoembryonic antigen (CEA); (**B**) basophil counts; (**C**) effector Vδ2 T cells. The lines within each box represent the median values, the boxes’ limits indicate the first and third quartiles, and the whiskers represent the smallest and largest values within 1.5 times the IQR from the first and third quartiles. *p*-values were determined using the Mann–Whitney–Wilcoxon test and *p* < 0.05 was considered significant.

**Figure 2 cells-13-01391-f002:**
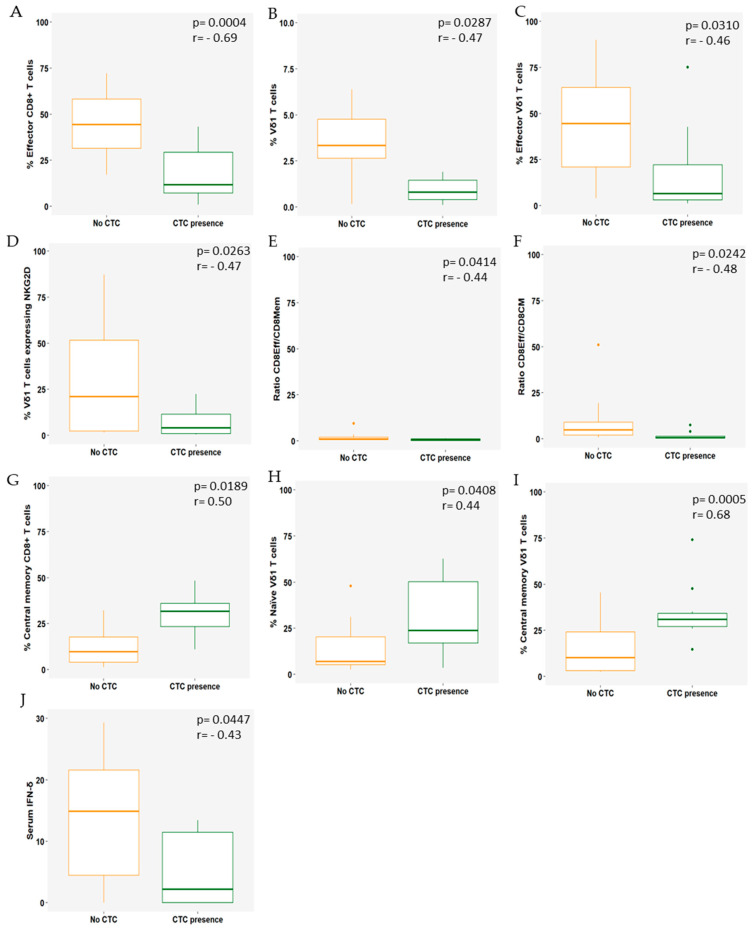
Significant correlations between CTCs (presence or absence at baseline with immune subsets and serum IFN−δ. (**A**) effector CD8+ T cells; (**B**) Vδ1 T cells; (**C**) effector Vδ1 T cells; (**D**) Vδ1 T cells expressing NKG2D; (**E**) ratio CD8Eff/CD8Mem; (**F**) ratio CD8Eff / CD8CM; (**G**) central memory CD8+ T cells; (**H**) naïve Vδ1 T cells; (**I**) central memory Vδ1 T cells; (**J**) serum IFN−δ. The absence of CTCs at baseline is represented as an orange boxplot and the presence of CTCs at baseline (≥1) is represented as a green boxplot. The lines within each box represent the median values, the boxes’ limits indicate the first and third quartiles, and the whiskers represent the smallest and largest values within 1.5 times the IQR from the first and third quartiles. *p*-values and r-coefficients were determined using point-biserial correlation and *p* < 0.05 was considered significant.

**Figure 3 cells-13-01391-f003:**
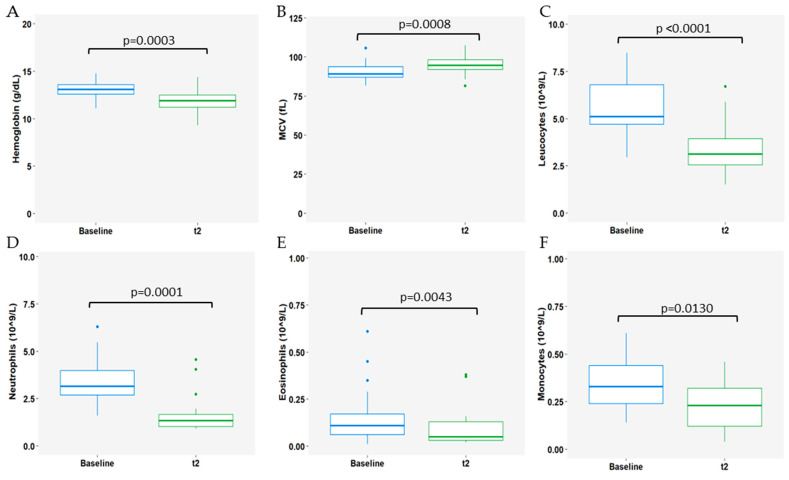
Impact of 3 months (t2) of CDK4/6i+ET on (**A**) hemoglobin levels (g/dL); (**B**) mean corpuscular volume (MCV; fL); (**C**) leucocyte counts (×10^9^/L); (**D**) neutrophil counts (×10^9^/L); (**E**) eosinophil counts (×10^9^/L); and (**F**) monocyte counts (×10^9^/L). Baseline is represented as a blue boxplot and t2 is represented as a green boxplot. The lines within each box represent the median values, the boxes’ limits indicate the first and third quartiles, and the whiskers represent the smallest and largest values within 1.5 times the IQR from the first and third quartiles. *p*-values were determined using the Wilcoxon test and *p* < 0.05 was considered significant.

**Figure 4 cells-13-01391-f004:**
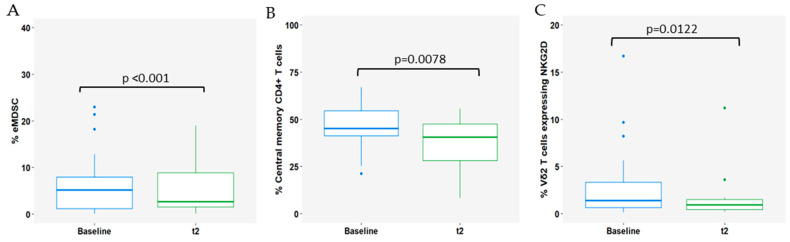
Impact of 3 months (t2) of CDK4/6i+ET on circulating immune cell subsets: (**A**) early-stage myeloid-derived suppressor cells (eMDSCs); (**B**) central memory CD4+ T cells; and (**C**) Vδ2 T cells expressing NKG2D. Baseline is represented as blue boxplot and t2 is represented as green boxplot. The lines within each box represent the median values, the boxes’ limits indicate the first and third quartiles, and the whiskers represent the smallest and largest values within 1.5 times the IQR from the first and third quartiles. *p*-values were determined using Wilcoxon test and *p* < 0.05 was considered significant.

**Figure 5 cells-13-01391-f005:**
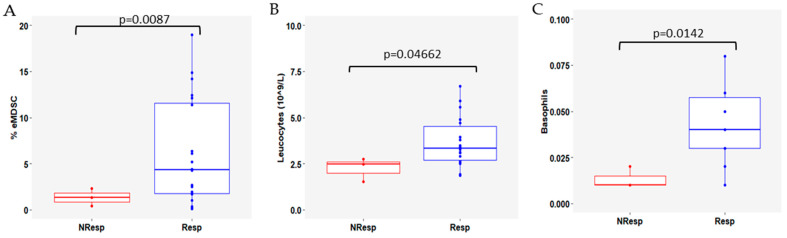
Comparison between Resp and NResp patients 3 months after the start of CDK4/6i+ET (t2) for (**A**) early-stage myeloid-derived suppressor cells (eMDSCs); (**B**) leucocyte counts (×10^9^/L); and (**C**) basophil counts. NResp patients are shown as red box plots and Resp patients are shown as blue box plots. The lines within each box represent the median values, the boxes’ limits indicate the first and third quartiles, and the whiskers represent the smallest and largest values within 1.5 times the IQR from the first and third quartiles. *p*-values were determined using the Mann–Whitney–Wilcoxon test and *p* < 0.05 was considered significant.

**Figure 6 cells-13-01391-f006:**
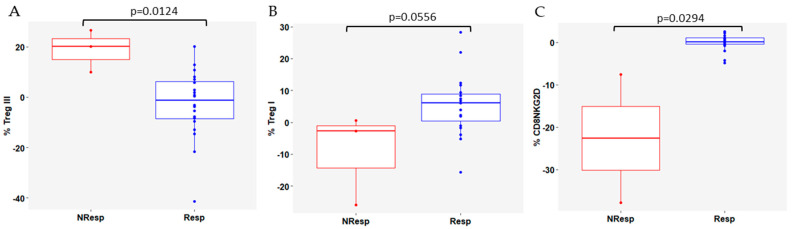
Variation in (**A**) Treg III; (**B**) Treg I; and (**C**) CD8+ T cells expressing NKG2D, between baseline and t2 in Resp and NResp patients. NResp patients are shown as red box plots and Resp patients are shown as blue box plots. The lines within each box represent the median values, the boxes’ limits indicate the first and third quartiles, and the whiskers represent the smallest and largest values within 1.5 times the IQR from the first and third quartiles. *p*-values were determined using the Mann-Whitney-Wilcoxon test and *p* < 0.05 was considered significant.

**Table 1 cells-13-01391-t001:** Immunophenotypic signature and role of various T-cell subtypes.

	Signature	Role
**T cell subtype**		
CD4 T cells [12]	CD3+ CD4+	T-helper cells - Belong to the adaptive T-cell immunity. - Support the cytotoxic T-cell differentiation and function.
CD8 T cells [13]	CD3+ CD8+	Cytotoxic T cells - Recognize and kill tumor cells by secreting effector cytokines (IFN-γ, TNF-α) and cytolytic molecules (granzyme B, perforin).
Treg cells [12,14,15,16,17]	CD3+ CD4+ FoxP3+	Regulatory T cells - Major subset of CD4+ T cells. - Regulate immune activity. - Suppress the anti-tumor immune effector response in TME.
γδ T-cells [18,19,20,21,22,23,24]		A subset of T cells that express a T-cell receptor composed of a γ and a δ chain. - Innate-like features. - Recognize and kill tumor cells by secreting effector cytokines (IFN-γ, TNF-α) and cytolytic molecules (granzyme B, perforin). Regulate the immune response through cytokine production, with both pro-inflammatory and regulatory effects.
Vδ1	CD3+ Vδ1+	Mostly found in mucosal and epithelial tissues.
Vδ2	CD3+ Vδ2+	Mostly found in circulation.
eMDSCs [14,25,26]	CD3− CD16− CD14− CD11b+	Early-stage myeloid-derived suppressor cells - Immature progenitors of Mo-MDSCs. A subset with the ability to suppress the anti-tumor effect of cytotoxic cells.
**Functional state**		
Naïve [27]	CD45RA+ CD27+	Naïve cells - Once activated, naïve cells can induce clonal expansion and effector and cytolytic function.
Eff [27]	CD45RA+ CD27−	Effector cells - Facilitate optimal immune responses against invading microbes and tumor antigens. - Cytotoxic effect.
EM [27,28,29]	CD45RA− CD27−	Effector memory cells - Major circulating memory cells. - Express effector molecules such as IFN-γ and perforin.
CM [28,29]	CD45RA− CD27+	Central memory cells - Mediate the persistent tumor immunity in the long term. - Higher proliferative capacity than EM. - Mainly express high levels of IL-2. - Have the ability to expand and differentiate to effector and effector memory subsets

CM, central memory; Eff, effector; EM, effector memory; eMDSCs, early-stage myeloid-derived suppressor cells; IL-2, interleukin-2; IFN-γ, interferon -γ; Mo-MDSCs, monocytic myeloid derived suppressor cells; TCR, T-cell receptor; TME, tumor microenvironment; TNF-α, tumor necrosis factor α.

**Table 2 cells-13-01391-t002:** Clinicopathological characteristics of the study cohort population.

	Responders		Non-Responders			Total		* p * -Value
	34 (77.3%)		10 (22.7%)			44 (100.0%)		
**Female, n (%)**	34 (100.0)		10 (100.0)			44 (100.0)		-
**Histology, n (%)**								
NST	25 (73.5)		7 (70.0)			32 (72.7)		0.850
Lobular	6 (17.6)		2 (20.0)			8 (18.2)		
MD	3 (8.8)		1 (10.0)			4 (9.1)		
**Grade, n (%)**								
1	1 (2.9)		1 (10.0)			2 (4.5)		0.686
2	17 (50.0)		6 (60.0)			23 (52.3)		
3	11 (32.4)		3 (30.0)			14 (31.8)		
MD	5 (14.7)		0 (0.0)			5 (11.4)		
**ER**								
Median (range)	100.0 (25.0–100.0)		90.0 (5.0–100.0)			95.0 (5.0–100.0)		0.174
MD, n (%)	10 (29.4)		2 (20.0)			12 (27.3)		
**PR**								
Median (range)	40.0 (0.0–100.0)		15.0 (1.0–100.0)			30.0 (0.0–100.0)		0.970
MD, n (%)	8 (23.5)		2 (20.0)			10 (22.7)		
**Ki67**								
Median	30.0 (5.0–75.0)		20.0 (5–40.0)			27.5 (5–75)		0.189
<20, n (%)	5 (14.7)		3 (30.0)			8 (18.2)		0.287
≥20, n (%)	21 (61.8)		5 (50.0)			26 (59.1)		
MD, n (%)	8 (23.5)		2 (20.0)			10 (22.7)		
**HER2, n (%)**								
0	23 (67.6)		6 (60.0)			29 (65.9)		0.848
1+ or 2+ ISH non-amplified	6 (17.6)		4 (40.0)			10 (22.7)		
HER2-negative (unclassified) †	5 (14.7)		0 (0.0)			5 (11.4)		
**DFS**								
De novo	7 (20.6)		1 (10.0)			8 (18.2)		0.747
≤24 mo	3 (8.8)		1 (10.0)			4 (9.1)		
>24 mo	24 (70.6)		8 (80.0)			32 (72.7)		
Median (range), mo *	85.6 (8.0–245.3)		80.69 (15.1–131.3)			84.4 (8.0–245.3)		0.064
**Symptomatic at metastatic disease diagnosis, n (%)**								
Yes	15 (44.1)		2 (20.0)			17 (38.6)		0.759
**Previous CT, n (%)**								
(Neo)Adjuvant setting	17 (50.0)		7 (70.0)			24 (54.5)		0.533
Metastatic setting	2 (5.9)		1 (10.0)			3 (6.8)		0.650
Clinicopathological characteristics at baseline								
**Age at baseline, years**								
Median (range)	57 (27–78)		50.5 (36–71)			56.5 (27–88)		0.305
**Menopausal status at baseline, n (%)**								
Pre- or peri-menopausal	12 (35.3)		4 (40.0)			16 (36.4)		0.786
Post-menopausal	22 (64.7)		6 (60.0)			28 (63.6)		
**Number of metastatic sites at baseline**								
Median (range)	1.5 (1.0–3.0)		1.0 (1.0–4.0)			1.0 (1.0–4.0)		0.226
1 metastatic site	17 (50.0)		9 (90.0)			26 (59.1)		0.024
≥2 metastatic sites	17 (50.0)		1 (10.0)			18 (40.9)		
**Metastatic sites at baseline, n (%)**								
Bone only	11 (32.4)		6 (60.0)			17 (38.6)		0.114
Bone	22 (64.7)		7 (70–0)			29 (65.9)		0.756
Lung	8 (23.5)		1 (10.0)			9 (20.5)		0.351
Liver	11 (32.4)		4 (40.0)			15 (34.1)		0.654
CNS	0 (0.0)		1 (10.0)			1 (2.3)		0.062
**CDK4/6i therapy line, n (%)**								
First	26 (76.5)		7 (70.0)			33 (75.0)		0.678
Second	8 (23.5)		3 (30.0)			11 (25.0)		
**ET used in combination with CDK4/6i, n (%)**								
AI	23 (67.6)		5 (50.0)			28 (63.6)		0.308
Fulvestrant	11 (32.4)		5 (50.0)			16 (36.4)		
OFS ^¥^	8 (23.5)		2 (20.0)			10 (22.7)		0.827
**CDK4/6i, n (%)**								
Ribociclib	13 (38.2)		3 (30.0)			16 (36.4)		0.605
Palbociclib	18 (52.9)		5 (50.0)			23 (52.3)		
Abemaciclib	3 (8.8)		2 (20.0)			5 (11.4)		
**BTA, n (%)**	16 (47.1)		6 (60.0)			22 (50.0)		0.295
**ECOG-PS at baseline, n (%)**								
0	22 (64.7)		7 (70.0)			29 (65.9)		0.640
≥1	7 (20.6)		1 (10.0)			8 (18.2)		
MD	5 (14.7)		2 (20.0)			6 (15.9)		

BTA, bone-targeted agent; CDK4/6i, cyclin-dependent kinase 4/6 inhibitor; CNS, central nervous system; CT, chemotherapy; DFS, disease-free survival; ECOG-PS, Eastern Cooperative Oncology Group Performance Status; ET, endocrine therapy; ER, estrogen receptor; HER2, human epidermal growth factor receptor 2; MD, missing data; mo, months; n, number; NST, non-special type; OFS, ovarian function suppression; PR, progesterone receptor. * for patients M0 at diagnosis; † HER2-negative not otherwise specified; ^¥^ OFS in combination with ET (either AI or fulvestrant) for pre-/peri-menopausal patients.

## Data Availability

For shared research data contact the corresponding authors.

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
