# Peer review of "Effect of Cyclin-Dependent Kinase 4/6 Inhibitors on Circulating Cells in Patients with Metastatic Breast Cancer"

_cells, 2024, doi:10.3390/cells13161391_

Round 1
Reviewer 1 Report
Comments and Suggestions for Authors
The manuscript by Lobo-Martins and distinguished colleagues addresses an important area in breast cancer therapy: The use of cyclin-dependent kinase 4/6 inhibitors (CDK4/6i) with endocrine therapy (ET). This study provides new insights into the immunomodulatory effects of CKD4/6i therapy on circulatory cells in patients with metastatic breast cancer. The study design consisted of 43 patients classified with ER+/HER2– advanced/metastatic breast cancer (ABC) treated with CDK4/6i and ET in either first or second line. Peripheral blood samples were collected before and 3 months after therapy, and patients were categorized as responders or non-responders. CDK4/6i therapy resulted in significant changes in hematological parameters, including decreased hemoglobin levels and reductions in specific immune cell subsets. Also noteworthy was the observation of circulating tumor cells (CTCs) with respect to immune cells, thus providing new insights into immune dysfunction and tumor dissemination. This paper highlights the importance of considering the immunomodulatory effects of CDK4/6i therapy in treating ER+/HER2- ABC patients.
The following comments are offered to the authors to improve the presentation of the study:
- - It is highly recommended that the ABC reference be changed to avoid confusion with ABC transporters mediating multidrug resistance in cancer.
- - For clarity, it would be helpful to always include CDK4/6i and ET - - to emphasize the combined therapy.
- - The transparent acknowledgment of the study’s limitations is excellent. Since heterogeneity is always an issue with most patient studies, the authors are encouraged to describe how they would address this in a future investigation.
- - Regarding the presentation of data in the figures, if space is an issue, the bar graphs could be condensed.
Author Response
Thank you for your time and effort in reviewing our manuscript. We appreciate your insightful comments and suggestions, which have significantly improved the quality of our paper.
- Comment 1
It is highly recommended that the ABC reference be changed to avoid confusion with ABC transporters mediating multidrug resistance in cancer.
Response: Thank you for this suggestion. We have changed the reference from "ABC" to "metastatic breast cancer (mBC)" throughout the manuscript to avoid any confusion.
- Comment 2
For clarity, it would be helpful to always include CDK4/6i and ET to emphasize the combined therapy.
Response: We appreciate this recommendation. Whenever applicable, we have opted for the designation "CDK4/6i+ET" to emphasize the combined therapy throughout the text.
- Comment 3
The transparent acknowledgment of the study’s limitations is excellent. Since heterogeneity is always an issue with most patient studies, the authors are encouraged to describe how they would address this in a future investigation.
Response: Thank you for your positive feedback. We recognize the challenges posed by heterogeneity in patient studies and appreciate the opportunity to elaborate on why, despite our efforts, heterogeneity was still present in our study and how we plan to address it in future investigations. We have updated the discussion section to reflect these considerations more thoroughly.
Strategies Already Implemented to Address Heterogeneity:
- Prospective Cohort Design: We had implemented a prospective cohort design to systematically collect data over time, providing a controlled environment to monitor changes and outcomes in patients receiving CDK4/6i and ET.
- Single Institution with Expanded Patient Referral: By accepting patients referred from other institutions, we increased our cohort size and enhanced statistical power within each subgroup, addressing heterogeneity to some extent.
- Coordinator Support for Adherence: Dedicated coordinators at our center assisted patients and physicians in adhering to study requirements, ensuring consistent data collection and follow-up.
Despite our efforts, there was still high heterogeneity, and here are some possible explanations for that:
- Real-World Evidence and Compliance: Real-world conditions inevitably introduce variability that can affect study outcomes. The COVID-19 pandemic, for instance, significantly impacted clinical trials worldwide, affecting patient recruitment, follow-up schedules, and adherence to protocols. Some patients were reluctant to return for blood collections, and many had altered assessment schedules, adding variability.
- Academic Initiative: As an academic initiative, our study benefits from a rigorous approach to data collection and analysis but faces limitations in resources and patient diversity compared to larger, multi-institutional trials, contributing to heterogeneity.
In future studies, we aim to implement additional strategies to monitor and improve compliance among patients and physicians, such as regular training sessions and follow-up reminders. By adopting these approaches, we aim to enhance the robustness and generalizability of our findings, contributing to more effective and tailored treatments for patients with metastatic breast cancer.
- Comment 4
Regarding the presentation of data in the figures, if space is an issue, the bar graphs could be condensed.
Response: We appreciate this suggestion. We have reviewed and condensed some figures to improve the flow and make the presentation more appealing to the reader without losing important information.
Reviewer 2 Report
Comments and Suggestions for Authors
The authors provide a fine-grained study on the effects of CDK4/6 inhibitor therapy on breast cancer patients. They specifically looked at the peripheral blood immune cell and circulating tumor cells profiles. The data showed correlation with response with specific blood cell populations. Some of the comments are as follows:
Table 2 with clinic pathological data might require some clarification. Did the authors test between multi-site metastasis vs single site? Similar with “ET used in combination with” group, since the number exceeds 43, does this mean a patient might have received as both AI and Fulvestrant? Was there any statistically significant difference between patients with multiple combination drug other than CDK4/6 inhibitor? Could they clarify why a total of 43 were chosen? Did they have most data available for them or CTC isolation was possible.
Could the authors elaborate on the lines 145-147? Are these scripts part of cellsearch? It would be helpful to link to the scripts or provide them as supplemental materials.
For line 157, did the author consider running the correlation with the number of CTCs instead of treating this as dichotomous. was regression not conducted due to low number of samples?
The authors could include the specific references for each subtype in table 1 instead of mentioning them at the top.
Reference from 29 is missing from the main text.
Line 164 mention R packages used if possible. When a specific cell type is compared, is the comparison between the specific % derived from flow cytometry data or some sort of a count/ml)?
Line 288, should be blue and green.
Author Response
Thank you for your time, detailed and constructive feedback. Your comments have been invaluable in refining our manuscript and enhancing the clarity and robustness of our findings.
- Comment 1
Table 2 with clinicopathological data might require some clarification. Did the authors test between multi-site metastasis vs single site? Similar with “ET used in combination with” group, since the number exceeds 43, does this mean a patient might have received both AI and Fulvestrant? Was there any statistically significant difference between patients with multiple combination drugs other than CDK4/6 inhibitor? Could they clarify why a total of 43 were chosen? Did they have most data available for them or CTC isolation was possible?
Response: Thank you for these insightful questions. We have provided clarifications and additional details in response to each part of your comment:
- Selection of 44 Patients:
The patients included in this analysis were part of the ONCODYNAMICS BioBanking (ODB) prospective cohort study, which is designed to support precision medicine in cancer by evaluating tumor clonal evolution, host immune response, and circulating biomarkers.
To ensure that this subgroup analysis was more robust and homogeneous, we restricted it to patients with the following criteria: ER+/HER2- metastatic breast cancer, treated with CDK4/6 inhibitors in combination with endocrine therapy in either the first- or second-line setting, with comprehensive clinical data available. These criteria were chosen to minimize variability and enhance the reliability of our findings. This approach leverages real-world evidence, reflecting the complexities and variations seen in clinical practice. By focusing on a well-defined subgroup, we aimed to provide more precise and actionable insights into the effects of CDK4/6 inhibitors in this patient population.
We have updated the Methods section 2.1. Study Design and Eligibility Criteria to explicitly state that the patients for this sub-analysis were selected from the ODB cohort. Additionally, we have created Supplementary Data S1, which details the specific inclusion and exclusion criteria used for the selection of patients for this sub-analysis. By including these details, we aim to provide a comprehensive understanding of the patient selection process and assure the reviewers that the methodology is sound and transparent.
Initially, there was an error in reporting the total number of patients. The correct total is 44 patients. We have updated Table 2 with the corrected number of patients and have redone all the statistical analyses accordingly. For the rest of the paper, the analysis was correctly done for all 44 patients. This clarification has been included in the revised manuscript.
- ET Used in Combination:
The number exceeding 100% in the "ET used in combination with" group is due to the addition of ovarian function suppression (OFS) for pre- and peri-menopausal patients, in combination with either AI or Fulvestrant. We have added a note on OFS in Table 2 and mentioned in the footnote that OFS is used in combination with AI or Fulvestrant for pre- and peri-menopausal patients. We hope this provides the necessary clarification.
- Testing Between Multi-Site Metastasis vs Single Site:
We have conducted an analysis to compare patients with multi-site metastasis (≥ 2 metastatic sites) versus those with a single site (1 metastatic site). The results of this analysis have been added to Table 2.
- Comment 2
Could the authors elaborate on the lines 145-147? Are these scripts part of cellsearch? It would be helpful to link to the scripts or provide them as supplemental materials.
Response: The authors appreciate this comment. The used scripts are not part of CellSearch, CellSearch’s CELLTRACKS ANALYZER II software program are entirely proprietary to Menarini Silicon Biosystems Inc. or its licensors.
The scripts, interface, procedures and operations used for image segmentation and cell classification were developed within the resources and expertise of group of authors, which consider it holds relevant innovation and industrial applicability and therefore have been analyzing intellectual property protection approaches. Due to the exposed, authors are protective and precautionary of detailed descriptions and disclosures on the subject, and therefore it was not further detailed in the manuscript.
- Comment 3
For line 157, did the author consider running the correlation with the number of CTCs instead of treating this as dichotomous. was regression not conducted due to low number of samples?
Response: Thank you for your question. A large number of patients (n=12) presented no circulating tumor cells (CTCs), making the calculation of Pearson correlation inadequate due to the excess of zero values. Initially, we considered dichotomizing the CTC counts into presence/absence and applying linear regression (or equivalently, performing a t-test). However, these approaches were not pursued as the distributions of the populations under comparison exhibited high skewness, which violates the assumptions of these parametric tests.
While comparing groups using the Mann-Whitney-Wilcoxon (MWW) test would be a non-parametric alternative, we ultimately decided not to do it since the primary objective of our study was to assess correlation rather than differences between groups. Therefore, the use of correlation-specific methods was deemed more appropriate for our analysis.
- Comment 4
The authors could include the specific references for each subtype in table 1 instead of mentioning them at the top.
Response: Thank you for your suggestion. We have updated Table 1 to include the specific references for each cell type directly within the table. This change provides clearer attribution and makes it easier for readers to locate the relevant references.
- Comment 5
Reference from 29 is missing from the main text.
Response: Thank you for bringing this to our attention. There was an issue with our reference management software, Zotero, which caused references beyond number 28 to be omitted from the main text. We have corrected this issue, and the references have been updated to include all 55 references in the revised manuscript. Reference 29 and all other references are now correctly cited in the main text.
- Comment 6
Line 164 mentions R packages used if possible. When a specific cell type is compared, is the comparison between the specific % derived from flow cytometry data or some sort of a count/ml?
Response: Thank you for your query. We have added the R packages used in the revised manuscript. Regarding the comparison of specific cell types, we have clarified in the manuscript that the comparison is based on the percentage derived from flow cytometry data. The phrase "the percentage of" has been added for clarity.
- Comment 7
Line 288, should be blue and green.
Response: Thank you for pointing this out. We have corrected the color notation in line 288 to "blue and green" in the revised manuscript.